# Feasibility of Predicting Surgical Duration in Endometriosis Using Numerical Multi-Scoring System of Endometriosis (NMS-E)

**DOI:** 10.3390/biomedicines12061267

**Published:** 2024-06-06

**Authors:** Masao Ichikawa, Tatsunori Shiraishi, Naofumi Okuda, Shigeru Matsuda, Kimihiko Nakao, Hanako Kaseki, Go Ichikawa, Shigeo Akira, Masafumi Toyoshima, Yoshimitu Kuwabara, Shunji Suzuki

**Affiliations:** 1Department of Obstetrics and Gynecology, Nippon Medical School, 1-1-5 Sendagi, Bunkyo, Tokyo 113-8602, Japan; m08029@nms.ac.jp (N.O.); m-shigeru@nms.ac.jp (S.M.); kimihiko@nms.ac.jp (K.N.); m-toyoshima@nms.ac.jp (M.T.); kuwa@nms.ac.jp (Y.K.); shun@nms.ac.jp (S.S.); 2Department of Obstetrics and Gynecology, Nippon Medical School Chiba Hokuso Hospital, 1715 Kamagari, Inzai, Chiba 270-1694, Japan; t-shiraishi@nms.ac.jp (T.S.); hanakow@nms.ac.jp (H.K.); go-ichikawa@nms.ac.jp (G.I.); 3Meirikai Tokyo Yamato Hospital, 36-3 Honcho Itabashi, Tokyo 173-0001, Japan; s-akira@nms.ac.jp

**Keywords:** endometriosis severity prediction, numerical multi-scoring system of endometriosis (NMS-E), preoperative diagnostic method, deep endometriosis, transvaginal ultrasound, pelvic examination, surgical duration estimation, laparoscopic surgery for endometriosis, r-ASRM classification

## Abstract

Background: Endometriosis is a multifaceted gynecological condition that poses diagnostic challenges and affects a significant number of women worldwide, leading to pain, infertility, and a reduction in patient quality of life (QoL). Traditional diagnostic methods, such as the revised American Society for Reproductive Medicine (r-ASRM) classification, have limitations, particularly in preoperative settings. The Numerical Multi-Scoring System of Endometriosis (NMS-E) has been proposed to address these shortcomings by providing a comprehensive preoperative diagnostic tool that integrates findings from pelvic examinations and transvaginal ultrasonography. Methods: This retrospective study aims to validate the effectiveness of the NMS-E in predicting surgical outcomes and correlating with the severity of endometriosis. Data from 111 patients at Nippon Medical School Hospital were analyzed to determine the correlation between NMS-E scores, including E-score—a severity indicator—traditional scoring systems, surgical duration, blood loss, and clinical symptoms. This study also examined the need to refine parameters for deep endometriosis within the NMS-E to enhance its predictive accuracy for disease severity. Results: The mean age of the patient cohort was 35.1 years, with the majority experiencing symptoms such as dysmenorrhea, dyspareunia, and chronic pelvic pain. A statistically significant positive correlation was observed between the NMS-E’s E-score and the severity of endometriosis, particularly in predicting surgical duration (Spearman correlation coefficient: 0.724, *p* < 0.01) and blood loss (coefficient: 0.400, *p* < 0.01). The NMS-E E-score also correlated strongly with the r-ASRM scores (coefficient: 0.758, *p* < 0.01), exhibiting a slightly more excellent predictive value for surgical duration than the r-ASRM scores alone. Refinements in the methodology for scoring endometriotic nodules in uterine conditions improved the predictive accuracy for surgical duration (coefficient: 0.752, *p* < 0.01). Conclusions: Our findings suggest that the NMS-E represents a valuable preoperative diagnostic tool for endometriosis, effectively correlating with the disease’s severity and surgical outcomes. Incorporating the NMS-E into clinical practice could significantly enhance the management of endometriosis by addressing current diagnostic limitations and guiding surgical planning.

## 1. Introduction

Endometriosis, characterized by the aberrant growth of endometrial-like tissue outside the uterine cavity, poses significant diagnostic and therapeutic challenges [1]. Its heterogeneous manifestations, including severe pain, infertility, and diverse lesion morphologies, necessitate a multifaceted diagnostic and management strategy [2,3].

A critical examination of current diagnostic and classification methods for endometriosis highlights a range of shortcomings. The revised American Society for Reproductive Medicine (r-ASRM) classification [4,5,6], which was modified in 1996 and is now widely known, utilizes a scoring system based on the size of endometriosis lesions and the extent of adhesions. This system classifies the disease into stages I to IV by accumulating points, with a maximum score of 150 points. While this feature proves to be very useful, the r-ASRM classification also presents significant limitations. Despite its widespread use, it falls short of accurately assessing deep endometriosis (DE) and lacks applicability as a preoperative diagnostic tool [3].

The Enzian score was initially published in 2005 as an independent postoperative assessment for DE [7]. It has evolved, especially with modifications in 2010 and 2011, to complement the r-ASRM classification by filling in the gaps related to DE [8]. The Enzian score, modeled after the TNM classification for cervical cancer, considers the tumorigenic nature of DE, categorizing lesions in the pouch of Douglas into A, B, and C sections based on location and size. It also describes adenomyosis and bladder endometriosis as ‘F’ (far). Nonetheless, these were not preoperative diagnostic methods. Recently, an evolved #Enzian classification has emerged as a preoperative diagnostic method achievable through transvaginal ultrasonography [9,10], primarily practiced in Europe. This method may face various challenges, including the technical difficulty of detecting small deep lesions via ultrasonography and unclear associations between lesions and pain, among others [3,11,12,13].

In 2023, the American Association of Gynecologic Laparoscopists (AAGL) 2021 Endometriosis Classification [14] was put forward, advancing the original intraoperative classification [15] by adopting transvaginal ultrasonography [10,14], akin to the ♯Enzian classification, enabling it to be used as a preoperative diagnostic method. Based on expert surveys, this approach assigns surgical complexity scores to each lesion, providing a singular indicator of the severity of endometriosis [14,15]. Despite its convenience and the availability of a supportive app, this method still faces technical difficulties in assessing peritoneal, tubal, and ureteral lesions via ultrasonography [11]. It may not clarify the relationship between lesions and pain either [13,16].

Other diagnostic methods with unique features have been proposed (such as the EFI (Endometriosis Fertility Index) [17,18]; an ultrasound mapping system [19]; EBDRECT—a preoperative score for accurately predicting rectosigmoid involvement in patients with endometriosis [20]; UBESS (ultrasound-based endometriosis staging system) [21]; and others [3,13]). Still, they have not secured a place as convenient first-line diagnostic tools, as they require complex evaluation or MRI, are sometimes not preoperative, or are not feasible for other reasons.

Pelvic examination remains superior for detecting pelvic pain. The Beecham classification [22] for endometriosis, adept at capturing early lesions, is now rarely practiced, highlighting the missing integration of pelvic examination findings in current diagnostics.

Considering the ideal first-line diagnostic method for endometriosis, various requirements emerge [3], but we believe the following four are crucial:A simple, objective, non-invasive method that captures early lesions and the diverse states of endometriosis, including their localization and spread.A scoring system to stratify severe cases and guide referrals to specialized facilities.An anatomically intuitive and easily shareable format akin to the TNM classification that facilitates information exchange between physicians and patients.A method capable of capturing temporal changes that is useful as an indicator for surgical, medicinal, recurrent, and infertility interventions.

To meet the first condition, the foundational examination methods should include pelvic examination and transvaginal ultrasonography. The second condition requires the technique to be scored. For the third, an easily shareable and anatomically illustrative format is needed. Lastly, the method should be non-invasive [23] and quick to execute by anyone anywhere to potentially fulfill the fourth condition.

The Numerical Multi-Scoring System of Endometriosis (NMS-E) was designed as a new comprehensive assessment tool for endometriosis, combining insights from pelvic examination and transvaginal ultrasonography. The full details of this system were published in the Japanese Society of Endometriosis journal in 2015 [24,25,26]. We have already reported on the outcomes related to the leading scores of the NMS-E, namely the adhesion score in 2020 [27] and pain score in 2023 [28]. Therefore, in this instance, we retrospectively investigated whether the E-score, a severity indicator in the NMS-E, actually correlates with the severity of endometriosis.

This study aims to address the gap in endometriosis diagnosis by evaluating the feasibility and efficacy of the NMS-E in predicting surgical duration and outcomes. By leveraging a retrospective analysis of patients treated for endometriosis at our institution, we seek to validate the NMS-E against traditional scoring systems and assess its potential to enhance surgical planning and patient management. Our hypothesis posits that the NMS-E can provide a more accurate reflection of disease severity, thereby improving preoperative predictions and surgical outcomes for patients with endometriosis.

## 2. Materials and Methods

### 2.1. Study Subjects

This diagnostic study used data from a previous study at Nippon Medical School Hospital. Of the 131 cases previously examined [27,28], 111 patients who underwent surgery at Nippon Medical School between 2012 and 2018 were included in the study and underwent preoperative transvaginal ultrasound and pelvic examination. The 20 cases that were excluded included 8 instances of large fibroid removal where the fibroids exceeded 3 cm, 6 cases where there was no complete resolution of Douglas pouch occlusion, 4 cases involving the resection of adenomyosis, and 2 cases of additional surgeries: one septectomy and one bladder repair due to bladder injury. MI, a physician with extensive experience in diagnosing and operating endometriosis, performed the examinations. This study was conducted in accordance with the ethical standards of the institutional review board and with the 1964 Helsinki declaration and its later amendments. Ethical approval for this study was obtained from the Ethics Committee of Nippon Medical School (Approval No. B-2020-261). The project identification code is B-2020-261, and the approval was granted on 3 March 2021. Informed consent was obtained from each patient who participated.

### 2.2. NMS-E Method

#### 2.2.1. NMS-E Layer Descriptions

The NMS-E stands for the Numerical Multi-Scoring System of Endometriosis, a preoperative, non-invasive diagnostic approach that leverages pelvic examination and transvaginal ultrasonography to assess endometriosis comprehensively. This method is structured into three distinct layers, as depicted in Figure 1, which describes the structure and application of the methodology. Appendix A consists of blank sheets intended for the reader’s use, and we have provided the NMS-E calculator Excel sheet in the Appendix A.

Physical Finding Map: This foundational layer organizes a wide array of endometriosis-related data collected during examinations into a visual and anatomical format. It uses a 3 × 3 grid system to map transvaginal ultrasound views of the uterus and ovaries from different perspectives and quantifies pain intensity in seven pelvic regions using the Numeric Rating Scale (NRS), as assessed during the pelvic examination.NMS-E Summary: Building on the Physical Finding Map, this layer condenses the detailed observations into a standardized summary formula, akin to the TNM classification used in oncology. It summarizes key findings, such as the size of endometriomas, the extent of adhesions, the intensity of pain, and identified uterine lesions, calculating a score for each. Some examples of NMS-E summaries are shown in Appendix B—NMS-E Examples 1–5.E-Score: The culmination of the NMS-E method, this layer translates the summarized data into a singular numeric value, representing the severity of endometriosis. The E-Score integrates critical elements derived from the previous layers—cyst, adhesion, pain, and uterine scores—to provide a comprehensive measure of disease severity.

#### 2.2.2. The Measurement and Recording Methods for the Four Conditions of Endometriosis

Endometrioma: Endometrioma is measured using the maximum diameter in transvaginal ultrasonography (to one decimal place, in cm). The findings are recorded in the central row’s left cell of the left 3 × 3 grid (corresponding to the right adnexal region of the patient) or the right cell (corresponding to the left adnexal region) (Figure 1’s left 3 × 3 grid). For multi-cystic conditions, the total maximum diameter is used. For non-endometriomas, the cyst type’s initial is prefixed before the size [Appendix B—NMS-E Example 3]. If tubal lesions are identified, their abbreviations (e.g., hydrosalpinx: h.s) are also recorded [Appendix B—NMS-E Example 4]. For summarization, the rounded-up value of the cyst’s maximum diameter is recorded as the cyst score for each side. Figure 1 shows a right endometrioma of 6.5 cm and a left endometrioma of 2.4 cm. For summarization, values are rounded up, resulting in 7/3. The maximum score for a single endometrioma is 5. Thus, the cyst score for this case is right 5 + left 3 = 8 points.Adhesion: Adhesion is measured at ten locations using transvaginal ultrasonography, with cross-sectional (five locations) and longitudinal (five locations) images of the uterus ovaries (left and middle 3 × 3 grids of Figure 1). Adhesions are assessed based on the presence or absence of the sliding sign at each location. The sliding sign is a method used to diagnose adhesions by checking if there is movement between the target organ and surrounding tissue when pressed with an ultrasound probe [29]. The presence of movement indicates no adhesions (−), while its absence indicates adhesions (+). For more details on the measurement method, refer to the adhesion score paper [27]. Measurement locations include the space between the right ovary and right pelvic wall (Rt. O-side); the space between the right ovary and uterus (Rt. O-Ut.); the space between both ovaries (Inter O-O); the space between the left ovary and uterus (Lt. O-Ut.); the space between the left ovary and left pelvic wall (Lt. O-Side); the upper (Upper ant.) and middle (Mid. ant.) parts of the anterior surfaces of the uterus; and the upper (Upper post.), middle (Mid.post.), and lower (Lower post.) parts of the posterior surfaces of the uterus. Figure 1 shows adhesions at four locations in the cross-sectional view and three in the longitudinal view, resulting in an adhesion score of 7/10. This value is directly added to the E-score.Pain: Pain is evaluated using the NRS (out of 10) based on pain induced by palpation during pelvic examination in seven pelvic regions centered around the uterine cervix: I. the right adnexal region, II. the right uterosacral ligament area, III. the anterior vaginal wall area, IV. the cervical area, V. the pouch of Douglas, VI. the left adnexal region, and VII. the left uterosacral ligament area. The values are recorded in the corresponding cells of the right 3 × 3 grid in Figure 1. The details of this mapping, including pain intensity in each region, are referenced in the pain score paper [28]. The highest point among the seven areas is the max pain score. In Figure 1, the highest point is 8 in the Douglas pouch area, making the patient’s pain score 8. This value is directly added to the E-score.Uterine Lesion: Uterine lesions are mainly evaluated using transvaginal ultrasonography. The assessed conditions include a retroverted uterus (R), endometriotic nodules (E), and adenomyosis (A). Uterine fibroids (M) are evaluated but not scored. All detected conditions, such as R, A, E, and M lesions, and their sizes (if applicable), are recorded in the central cell of the middle row of the central grid or the anatomically corresponding cell. Endometriotic nodules (E) are depicted as hypoechogenic lesions similar to adenomyosis outside the uterus (lesions larger than 1 cm in diameter are defined as E in this assessment) [10]. E lesions detected as nodules during pelvic examination are also marked in the corresponding cell on the right grid. The definition of a retroverted uterus (R) is when the angle formed by the cervical and uterine body axes is less than 180 degrees posteriorly [30]. Adenomyosis (A) appears on transvaginal ultrasound as a heterogeneously enlarged uterus with myometrial cysts, asymmetric myometrial thickening, poorly defined areas of echogenicity, etc. [10]. Figure 1 displays a 2.6 cm endometriotic nodule located at the center of the posterior uterine surface, along with adenomyosis. When summarizing, these lesions are shown as A and E, and when scoring, each lesion is given 3 points, giving a total of 6 points. However, in some of the data of this research, E lesions between 1 cm and 2 cm are scored as 3 points, between 2 cm and 3 cm as 6 points, and larger than 3 cm as 10 points.Rare-site Endometriosis: Rare-site endometriosis is treated separately from the abovementioned four states. The diagnostic methods vary depending on the location of the lesion. Lesions considered for rare-site endometriosis evaluation include intestinal endometriosis, bladder endometriosis, ureteral endometriosis, vaginal endometriosis, cutaneous endometriosis, etc. If these are observed, they are added to the end of the NMS-E as an additional notation (Appendix B—Example 5). In scoring, rare-site endometriosis is tentatively assigned 10 points.

The transvaginal ultrasonography device used in this study was the Voluson E8 (GE Healthcare, Tokyo, Japan).

#### 2.2.3. How to Calculate the E-Score

The E-score quantifies the severity of endometriosis by combining scores from the following four categories detailed in the NMS-E summary and the score for rare-site endometriosis.

Cyst Score: Summed from endometriomas on both ovaries. Cysts up to 5 cm are scored at their rounded-up size; cysts over 5 cm are capped at 5 points. Maximum combined score: 10 points (range: 0–10). Fallopian tube lesions are separately scored as 3 points each (additional range: 0–6).Adhesion Score: Equals the number of positive adhesion sites (range: 0–10).Pain Score: Derived from the highest value from seven pain sites on the pain map (range: 0–10).Uterine Score: Each lesion scores 3 points, multiplied by the number of lesions. Conditions include a retroverted uterus (R), endometrial nodules (E) of ≥1 cm, and adenomyosis (A). Uterine fibroids (M) are noted but not scored (range: 0–9).

Rare-Site Endometriosis: If detected, scored as 10 points.

The total of these scores constitutes the E-score. For example, consider Figure 1, where the ovarian endometriosis scores are 7 and 3 for the right ovary and left ovary, respectively. With the maximum score for a single cyst being 5, a combined cyst score of 8 is yielded. Adhesion scores are based on detected sites (7 out of a possible 10), and the highest pain observed scores an 8. The uterine score, accounting for two lesions, is 6 (3 × 2). Without rare-site endometriosis, the total E-score is 29. 

For those who want to know how it is actually calculated, please refer to the accompanying NMS-E (E-score) calculator Excel sheet, which can be found in the Appendix A.

### 2.3. Statistics

Data normality was assessed using the Shapiro–Wilk test, conducted with R statistical software version 4.4.0. The Shapiro–Wilk test revealed that the E-scores did not follow a normal distribution, with a W statistic of 0.97205 and a *p*-value of 0.01966, which is below the conventional alpha level of 0.05.

Statistical analysis, including the correlation between scores such as the E-score and operation time or the VAS value, was performed using Spearman’s rank correlation coefficient, calculated with RANK.EQ and the CORREL function in Microsoft Excel (Version 16.77.1, Microsoft Corp., Redmond, WA, USA). A *p*-value of 0.05 or less was considered to indicate statistical significance.

## 3. Results

### 3.1. Demographics, Clinical Presentation, and Surgical Interventions in Endometriosis Patients

In the observed cohort of 111 patients, the mean age was 35.1 years, ranging from 23 to 51. The mean Body Mass Index (BMI) was 20.6 kg/m^2^, ranging from 17.2 to 28.5 kg/m^2^. The average parity was 0.3, with a range between 0 and 2. Out of 111 individuals, 92 (82.9%) were nulliparous (Table 1).

The clinical symptoms evaluated included dysmeno0rrhea with a mean Visual Analogue Scale (VAS) score of 6.8, dyspareunia with a mean VAS score of 3.9, dyschezia with a mean score of 3.1, and chronic pelvic pain with a mean score of 2.2. The mean duration of the operation was 181.4 min, ranging from 51 to 421 min, and the mean blood loss was 65.7 mL, ranging from 0 to 500 mL (Table 1).

Regarding endometrioma, 0.9% of patients had none, 45.0% had unilateral endometrioma, and 54.1% had bilateral endometrioma. Specific surgical interventions included unilateral cystectomy in 38.7% of patients and bilateral cystectomy in 27.0% of patients. The pouch of Douglas was reported as usual in 28.8% of patients, with partial obstruction in another 28.8% and complete obstruction in 42.3%. In all cases with partial or complete obstruction of the Douglas pouch, a full resolution of the obstruction was achieved.

Other conditions identified were adenomyosis in 24.3% of the cohort, with no patient undergoing resection for large or multiple myomas. A small proportion of patients underwent resection for vaginal endometriosis (2.7%), umbilical endometriosis (0.9%), and inguinal endometriosis (0.9%). Myomas were present in 16.2% of patients, with 8.1% undergoing resection for small myomas.

#### 3.1.1. Assessment of Endometriosis Severity Using r-ASRM and E-Score Metrics

In assessing endometriosis severity among the study population (*n* = 111), the mean revised American Society for Reproductive Medicine (r-ASRM) score was 69.4, ranging from 10 to 150 (Table 2). The distribution of severity stages indicated that the majority of the patients were classified with stage IV endometriosis (71.9%), followed by stage III (26.3%), while stages II and I were notably less common, representing 0.9% and 0.0% of the cases, respectively.

The mean E-score, which provides a quantified measure of endometriosis severity, was calculated at 20.32, within a range of 6 to 36. This composite score integrates individual metrics for cysts, adhesions, pain, and uterine abnormalities, with the mean score recorded as 6.32 for cysts, 3.96 for adhesions, 6.07 for pain, and 3.51 for uterine factors. The prevalence of specific uterine conditions was substantial, with endometriotic nodules (E) present in 52.2% of the patients, a retroverted uterus (R) present in 39.6% of the patients, and adenomyosis (A) present in 24.3% of the patients.

A Rare score assigned for less common sites of endometriosis was applied to five patients (4.5% of the study cohort), illustrating the occurrence of rare-site disease.

#### 3.1.2. Correlation of Endometriosis Scoring Systems with Surgical Duration and Blood Loss

The analysis presented in Figure 2a and Table 3 shows a correlation coefficient of 0.758 for the E-score and the r-ASRM score, indicating a substantial association between these two measures of endometriosis severity. The E-score also demonstrated a slightly stronger correlation with surgical duration, with a coefficient of 0.724 (Figure 2b). This was higher than the correlation between r-ASRM scores and surgical duration, which was 0.700. The adhesion score alone showed a notable correlation with surgical duration, with a coefficient of 0.640, and the uterine score followed closely with a coefficient of 0.491, suggesting these factors significantly influence the length of surgery (Table 3). The correlation between E-score and blood loss was the highest among the scores, with a coefficient of 0.400.

Following this analysis, the relationship between the E-score and clinical symptoms of endometriosis was further examined (Appendix A). Overall, the E-score hardly showed a significant correlation with these symptoms, indicating that while the E-score is predictive of surgical parameters, it may not correlate strongly with the clinical manifestations of the disease.

### 3.2. Refinement of Endometriotic Nodules Scoring and Its Impact on Surgical Duration Prediction in Endometriosis Treatment

Building on the data from Table 3, it was evident that the adhesion score contributed most significantly to the surgical score, followed by the uterine scores. This subsection explores how each element of the uterine score impacts operation time and whether modifying specific parameters improves the prediction of surgical duration (Table 4). Centered on the adhesion score, adding adenomyosis (A) as part of the uterine score slightly decreased the correlation with operation time (e.g., from 0.640 to 0.606). The addition of retroverted uterus (R lesion) showed the same change (e.g., from 0.640 to 0.618), while incorporating endometriotic nodules (E lesion) significantly improved the correlation (e.g., from 0.640 to 0.740). Given the relatively low score typically allocated to E lesions (3 points), employing a modified score that increases with lesion size (namely, assigning 3 points for lesions over 1 cm but under 2 cm, 6 points for lesions over 2 cm but under 3 cm, and 10 points for lesions over 3 cm) demonstrated an even stronger correlation (from 0.640 to 0.761). Finally, when adjusted to include these modified scores for E lesions, the initial E-score showed an improved correlation coefficient of up to 0.752.

## 4. Discussion

This study established that the E-Score of the NMS-E, a preoperative diagnostic indicator for assessing endometriosis’s severity that strongly correlates with the widely utilized r-ASRM score derived from surgical findings, with a correlation coefficient of 0.758. It also revealed a more substantial association with the duration of surgery (0.724) compared to the r-ASRM score (0.700). Furthermore, regarding blood loss, the E-Score demonstrated a higher correlation than the r-ASRM (0.400 vs. 0.328). These findings imply that the E-Score could be an equally or more precise predictor of endometriosis severity before surgery than the r-ASRM score.

Considering the predictive power of the E-Score for surgical duration, this study found that the expected surgery time based on the E-score can be determined by the following equation: y = 6.2995x + 49.459. For instance, an E-score of 19 points predicts a surgical time of 175 min. It should be noted that this equation might differ from one facility to another or from one surgeon to another, and even for the same surgeon, it could evolve with increasing experience. Nonetheless, with the accumulation of sufficient data, the precision of these predictions is expected to be enhanced, thereby facilitating strategic planning and readiness for surgeries within the institution.

Currently, beginning with #ENZIAN [31] and AAGL2021 [14], there is an active effort to estimate the severity of conditions including deep endometriosis using non-invasive preoperative diagnostic methods [21,32]. However, as of now, there have been no reports of attempts to predict surgical time using such diagnostic methods. It is expected that such reports will emerge in the future. Therefore, it will be necessary to evaluate which method provides more accurate predictions of surgical time: the NMS-E’s E-score or other methods.

Looking at individual scores, the adhesion score showed the highest correlation with surgical duration, scoring 0.640. This suggests that the strength of adhesions is the most significant factor determining surgical time or complexity. Following this, the uterine score, at 0.491, also shows a significant correlation. There is a perception that the excision of deep endometriosis (DE) lesions, which are identical to the uterine score’s ‘E’ and could cause adhesions, is time-consuming. In reality, at the AAGL 2021, it was reported that the expert evaluation for surgical complexity related to large deep endometriosis (DE) lesions (over 3 cm) almost scored the most points, between 9 and 10 out of 10 [15]. Next in line are the pain score and cyst score, respectively. It was unexpected that the time taken for cystectomy had a minor impact on the overall surgical time in this study. One reason for this could be that the majority of the subjects (71.9%) were patients with severe conditions at stage IV, suggesting that adhesions and deep endometriosis, more common in severe cases, could have been dictating the surgical duration. The cyst score might become a more significant determining factor in less severe conditions.

The significant correlation between the NMS-E E-score and surgical duration underscores the utility of the NMS-E as a predictive tool in clinical settings. This finding is particularly relevant for the surgical management of endometriosis, where estimating procedure length can aid in resource allocation and patient counseling. This feature indicates that the NMS-E meets the requirements sought in preoperative diagnostics for endometriosis [3].

In this study, the E-score of the NMS-E showed a high correlation with the r-ASRM score. Here, we want to consider the reason for this. The NMS-E was created to enable a non-invasive implementation of the r-ASRM, an intraoperative diagnostic method. Moreover, it incorporates elements of deep endometriosis, such as pain assessment, which are weaknesses of the r-ASRM [3]. Therefore, the correlation of these two diagnostic methods is not coincidental but by design. Here is a detailed explanation. The r-ASRM score is graded out of a total of 150 points: peritoneal lesions—6 points, endometriomas—40 points, posterior cul-de-sac obliteration—40 points, ovarian adhesions—32 points, and tubal adhesions—32 points [5]. On the other hand, the NMS-E grades out of approximately 40 points: endometriomas—10 points, ovarian adhesions—10 points, pain—10 points, and uterine lesions—9 points (3 points × 3) (in practice, the score limit is not fixed as additional points are given for tubal diseases and rare-site endometriosis) [24]. Comparing the elements of both diagnostic methods, the evaluation of the ovaries is nearly the same, and adhesions are common for the ovaries, with some overlap for posterior cul-de-sac obliteration (adhesion). Regarding pain, it is known that the areas with solid pain are around the cul-de-sac, and cases with posterior cul-de-sac obliteration significantly have higher pain scores [28]. Therefore, it can be said that the r-ASRM’s posterior cul-de-sac obliteration lesions and the NMS-E’s pain score share some common elements. This means the three main aspects of both diagnostic methods look almost identical. One significant difference is the evaluation of the fallopian tubes. Regular fallopian tubes are rarely visible upon transvaginal ultrasonography [11], so their adhesions are also unknown. Therefore, only when tubal enlargement is observed in the NMS-E is it graded with 3 points [24]. Another difference is that the NMS-E has a uterine score, evaluating deep lesions such as endometriotic nodules (in the r-ASRM, deep lesions are usually rated up to 6 points for peritoneal lesions, which is low compared to other items). The scoring of each element in the NMS-E is set to about 1/4 of each item’s score in the r-ASRM. For these reasons, the r-ASRM and NMS-E could show a high correlation.

However, even if the NMS-E and r-ASRM were diagnostic methods that encompass similar elements, another question might arise. Is it somewhat inconsistent to compare the NMS-E, a preoperative diagnostic method, with the r-ASRM classification, an intraoperative and different diagnostic method? Most elements of the preoperative diagnostic method, the NMS-E, can be confirmed intraoperatively. The E-score confirmed during surgery is referred to as the intraoperative E-score (iE-score). The confirmation methods are explained in Appendix A. The correlation coefficient for the correlation between the E-score and the iE-score was 0.932, as shown in Appendix A and Appendix A. Additionally, the correlation between the iE-score and the r-ASRM score was 0.803, and the correlation with surgical time was 0.712. Even in the iNMS-E, which assigns a higher score to E lesions, an improvement in the prediction rate similar to that of the NMS-E was observed, as noted in Appendix A. These results are considered to bridge the gap between the preoperative examination method of the NMS-E and the intraoperative examination method of the r-ASRM.

For preoperative diagnostic methods other than the NMS-E, the #Enzian [9] and the 2021 AAGL classifications [14,15] have recently gained attention [31]. It could also be possible to predict surgery times using these scores. However, attempts to predict surgical duration using these methods are not common, and there are only reports of attempts to predict surgery times using the traditional Enzian classification [33]. Therefore, it is unclear how accurately they can predict surgery times. Nevertheless, even if they could, we still believe the NMS-E has several advantages. One of them is the adhesion score, as mentioned above. The adhesion score, which has already been shown to diagnose the temporal change in the strength of postoperative adhesions and can be an indicator of infertility [27], is a unique score of the NMS-E that quantitatively measures adhesion strength out of 10 points and is unparalleled. Moreover, it has been demonstrated that it is a significant factor in determining surgery times. This is why we believe the NMS-E has superior surgery time prediction capabilities compared to other preoperative diagnostic methods. The base for the adhesion score measurement is also in the r-ASRM. In the r-ASRM score, the degree of solid adhesions around the ovary is classified into no adhesion, <1/3, 1/3 < < 2/3, or >2/3, and points are allocated to each adhesion state: 0 points, 4 points, 8 points, and 16 points, respectively [5].

In the NMS-E, it is assumed that the enlarged ovaries are placed within an inverted tetrahedron, and the presence of adhesions is evaluated on four surfaces: the ovarian surface (Inter O-O), the uterine surface (Lt O-Ut), the sidewall surface (Lt O-Side), and the upper surface (usually without adhesions). The loss of mobility on each surface is considered as the presence of adhesion. Thus, adhesions on one surface represent 1/4 coverage of adhesions, corresponding to less than 1/3 of adhesions in the r-ASRM. Adhesions on two surfaces represent 2/4 coverage, corresponding to between 1/3 and 2/3 in the r-ASRM, and adhesions on three surfaces represent 3/4 coverage, corresponding to more than 2/3 in the r-ASRM. In the adhesion score, 1, 2, or 3 points are assigned, respectively. This ingenuity has led to the adhesion score of the NMS-E not only correlating with the adhesion score of r-ASRM but also with the surgical duration.

Another significant advantage of the NMS-E is the existence of the pain score, derived from pelvic examination. Most endometriosis diagnostic methods do not include the assessment of pelvic examination. Dyspareunia is one of the critical indicators for deciding whether to perform surgery for endometriosis. There is no better method than a pelvic examination to detect such localized pain. Transvaginal ultrasonography accurately diagnoses deep lesions, but reports are scarce on strategies that can simultaneously assess the pain they induce [34]. Moreover, they are not comprehensive preoperative diagnostic methods for endometriosis. The NMS-E establishes a system that successfully integrates pelvic examination findings and transvaginal ultrasonography imaging using the pain score. The pain score has been shown to correlate most strongly with dyspareunia. These features make the NMS-E an unparalleled diagnostic method of great value, capable of predicting not only surgical duration but also the activity of deep lesions.

The limitation of this study is that only one examiner used this method. As a result, we obtained consistent data, but the possibility of bias is fully considered. Therefore, to prove that this method is universal, it is necessary to check the reproducibility of these data among many examiners and facilities and confirm its effectiveness.

Another problem is the small number of study cases. The current study used data from 111 cases, handled by a single operator for data standardization. Since endometriosis is a disease showing various pathologies, many confounding factors exist. Therefore, in the future, it is necessary to increase the number of cases further and make adjustments through matching and stratification.

Another significant issue in this study is the difficulty in determining the optimal weighting for each disease. In the NMS-E, in addition to the central four lesions, there are many parameters. Significant big data and effort are necessary to find optimal solutions for all of them. However, as mentioned before, since the NMS-E is somewhat based on the r-ASRM score weighting, there may not be such a significant empirical discrepancy. Nevertheless, to solve this problem relatively quickly, the way of scoring of the 2021 AAGL classification is a good reference [15]. For this classification, a survey was conducted on approximately 30 endometriosis expert physicians, and the complexity of each lesion was scored. The allocation of points to each lesion was determined based on the results. For example, complete cul-de-sac obliteration scored 9 points, endometriomas over 3 cm scored 7 points, ureteral endometriosis scored 6–9 points, and intestinal endometriosis over 3 cm scored 8 points. Coincidentally, the scoring is close to the NMS-E, where each element is nearly expressed out of a perfect score of 10 points. Also, in this and #Enzian classifications, the score jumps when the lesion size exceeds 3 cm. In our study, the correlation with surgical time improved when we also increased the scoring for the endometriotic nodules based on size (Table 4). It is necessary to consolidate this information in the future and make fine adjustments to the scoring in the NMS-E.

The clinical significance of the NMS-E is fourfold. First, using the E-score enables the preoperative identification of patients with severe endometriosis. Surgery for severe endometriosis often requires special procedures such as a complete opening of the obliterated cul-de-sac or shaving of intestinal endometriosis [35]. Therefore, patients with severe endometriosis should be operated on by experienced surgeons or at specialized facilities. However, until now, the severity of endometriosis was not apparent preoperatively, which might have led to inadequate triage and lost opportunities for adequate surgery for some patients. The use of the E-score can avoid such situations. Second, it enables the accurate prediction of surgery time, allowing for efficient operation room management, which is essential for hospital management and medical economics. Predicting surgery time for endometriosis, which can present complex conditions, was particularly challenging. This could lead to complex cases being scheduled late in the afternoon with a short expected surgery time, resulting in complications such as bowel injuries that require cooperation from other departments. On the other hand, if a case is considered mild based on the NMS-E preoperatively, it might be possible to plan more than three surgeries in one day, including that surgery. Third, the NMS-E summary facilitates sharing information about patient conditions among physicians, not just the severity. Although it may initially seem confusing, as seen in some examples in Appendix B, one can grasp the overall picture of endometriosis at a glance once accustomed. Finally, the Physical Finding Map allows for an understanding of the local conditions and the whole picture of endometriosis. This is particularly important as preoperative information. Based on this information, decisions can be made about removing a lesion and the lesion’s severity based on the location of the etiology and its activity (pain score). The Physical Finding Map becomes an indicator when planning surgical strategies. The NMS-E is a non-invasive preoperative diagnostic method that can be easily used with pelvic examination and transvaginal ultrasonography. With the above features, it realizes the ideal endometriosis diagnostic method initially proposed.

As a future research direction, it is necessary to validate the NMS-E further in more extensive and diverse populations to determine the effectiveness of this comprehensive preoperative diagnostic method for endometriosis, which incorporates numerous variables. Moving forward, collaborative research with multiple physicians and facilities is planned, and individual evaluations of various parameters, such as score limits and allocations, will also be conducted. Furthermore, developing non-invasive infertility prediction using the NMS-E is a crucial issue. For this purpose, it is necessary to add the critical information missing in the NMS-E. This information pertains to the patency of the fallopian tubes. However, it is impossible to diagnose the patency of the fallopian tubes solely based on the imaging information derived from transvaginal ultrasonography. Therefore, by combining tests such as the Tubal Insufflation Test (Rubin’s Test), saline-infused hysterosonogram (SIH), Hysterosalpingo-Contrast Sonography (HyCoSy) [36], or Hysterosalpingography (HSG), it may be possible to achieve the objective.

## 5. Conclusions

The NMS-E has been demonstrated to be a robust preoperative diagnostic tool, providing a strong correlation with the r-ASRM score and surgical outcomes, particularly in predicting surgical duration. Our study confirms that the E-score component of the NMS-E is a valuable predictor for endometriosis severity, showing a significant correlation with surgical duration and outperforming the r-ASRM in this respect.

Furthermore, refinements in scoring methodology, particularly concerning endometriotic nodules, have enhanced the predictive accuracy for surgical duration, potentially allowing for better preoperative planning and patient management. The NMS-E’s comprehensive approach, integrating findings from pelvic examinations and transvaginal ultrasonography, presents a multifaceted picture of endometriosis which can inform and improve clinical decision making and patient referrals.

Despite its effectiveness, it is recognized that the adoption of the NMS-E system requires validation across a broader spectrum of clinicians and institutions to ensure its universal applicability and reliability. Additionally, considering the NMS-E’s potential in surgical planning, further research could explore its predictive power for other surgical outcomes and the development of complementary tools to facilitate its use in clinical practice.

By addressing current diagnostic challenges in endometriosis, the NMS-E stands to significantly contribute to the optimization of patient care, not only enhancing the management of the disease but also the quality of life for those affected.

## Figures and Tables

**Figure 1 biomedicines-12-01267-f001:**
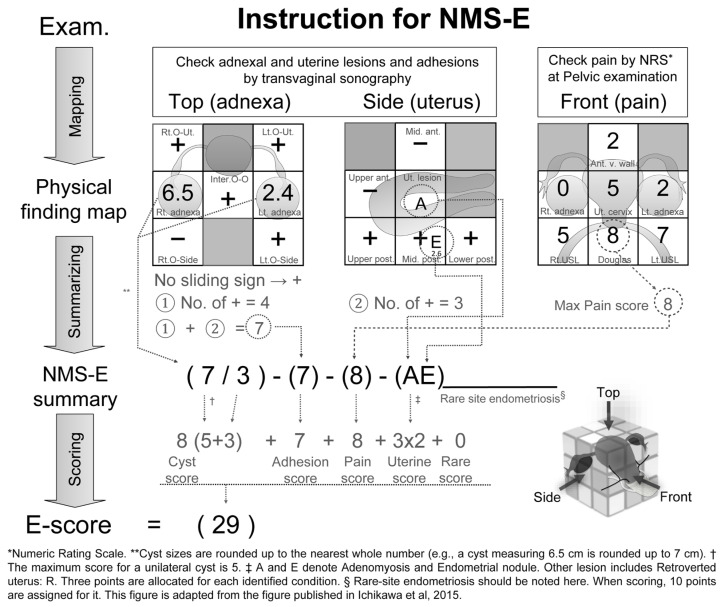
Instructions for NMS-E assessment sheet. The NMS-E assessment sheet offers a streamlined diagnostic approach for endometriosis, using a three-tiered structure: (1) a Physical Finding Map for a visual representation of findings; (2) the NMS-E summary, which summarizes data into a linear formula; and (3) the E-score, quantifying disease severity. Scores for ovarian endometriomas, adhesions, pain, and uterine lesions are totaled for the E-score. Rare-site endometriosis and conditions like tubal hydrosalpinx or pyosalpinx are also scored. The sheet visualizes and quantifies complex clinical data, facilitating a comprehensive understanding of endometriosis severity [24].

**Figure 2 biomedicines-12-01267-f002:**
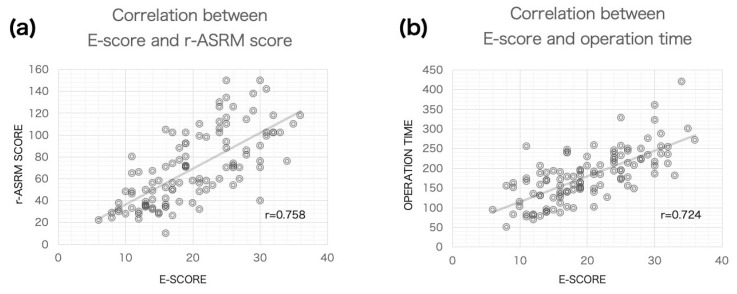
An analysis of the correlation of the E-score with the r-ASRM score and operation time. (**a**) This figure illustrates the correlation between the E-score and the r-ASRM score. (**b**) This figure shows the correlation between the E-score and operation time. Both figures are based on data from 111 subjects. The diagonal line in each graph represents the regression line.

**Table 1 biomedicines-12-01267-t001:** Patient characteristics and surgeries.

Patient Characteristics and Surgeries (*n* = 111)			Value
Mean Age (years old) [Min–Max]			35.1 (23–51)
Mean BMI (kg/m^2^) [Min–Max]			20.6 (17.2–28.5)
Mean Parity [Min–Max]			0.3 (0–2)
Number of Patients with Previous Medical Treatment for Endometriosis (*n*, %)			53 (47.7)
Number of Patients with Previous Surgery for Endometriosis (*n*, %)			11 (9.9)
Clinical symptoms			
	Dysmenorrhoea (*n* = 102 *). Mean VAS value		6.8
	Dyspareunia (*n* = 94 **). Mean VAS value		3.9
	Dyschezia (*n* = 102 *). Mean VAS value		3.1
	Chronic pelvic pain (*n* = 102 *). Mean VAS value		2.2
Mean operation time (min) [Min–Max]			181.4 (51–421)
Mean blood loss (mL) [Min–Max]			65.7 (0–500)
Endometrioma			
	None (*n*, %)		1 (0.9)
	Unilateral endometrioma (*n*, %)		50(45.0)
		Unilateral salpingo-oophorectomy (USO) (*n*, %)	4(3.6)
		Unilateral cystectomy (*n*, %)	43 (38.7)
		Ablation (*n*, %)	2 (1.8)
		others ^†^ (*n*, %)	1 (0.9)
	Bilateral endometrioma *(n*, %)		60 (54.1)
		USO + unilateral cystectomy (*n*, %)	8 (7.2)
		Bilateral cystectomy (*n*, %)	30 (27.0)
		Unilateral cystectomy + ablation (*n*, %)	19 (17.1)
		Others ^‡^ (*n*, %)	3 (2.7)
Douglas’ Pouch			
	Normal (*n*, %)		32 (28.8)
	Partial obstruction (*n*, %)		32 (28.8)
		Complete resolution of partial obstruction (*n*, %)	32 (28.8)
	Complete obstruction (*n*, %)		47 (42.3)
		Complete resolution of complete obstruction (*n*, %)	47 (42.3)
Other conditions			
	Adenomyosis (*n*, %)		27 (24.3)
		No resection (*n*, %)	27 (24.3)
		Resection of large or multiple adenomyosis	0 (0.0)
	Vaginal endometriosis resection (*n*, %)		3 (2.7)
	Umbilical endometriosis resection (*n*, %)		1 (0.9)
	Inguinal endometriosis resection (*n*, %)		1 (0.9)
	Myoma (*n*, %)		18 (16.2)
		Resection of small myoma (*n*, %)	9 (8.1)
		Resection of large or multiple myomas (*n*, %)	0 (0.0)
		No resection (*n*, %)	9 (8.1)
	Other surgeries ^§^ (*n*, %)		2 (1.8)

* Due to the preoperative VAS scores being unknown for 9 individuals, the data was based on 102 patients. ** For 9 individuals, the preoperative VAS scores were unknown, and 8 had not been sexually active for some time, thus the data was based on 94 patients. ^†^ This includes alcohol fixation and aspiration of cystic contents. ^‡^ This includes bilateral heat ablation. ^§^ This includes bartholinic cystectomy, and endometrial polypectomy. These durations have been subtracted from the total surgical time.

**Table 2 biomedicines-12-01267-t002:** Endometriosis severity in r-ASRM and NMS-E in endometriosis patients.

Scores (*n* = 111)			Value
Mean r-ASRM * score [Min–Max]			69.4 (10–150)
	I (1–5)	(*n*, %)	0 (0.0)
	II (6–15)	(*n*, %)	1 (0.9)
	III (16–40)	(*n*, %)	29 (26.3)
	IV (>41)	(*n*, %)	81 (71.9)
Mean E-score [Min–Max]			20.32 (6–36)
	Mean Cyst score [Min–Max]		6.32 (0–12)
	Mean Adhesion score [Min–Max]		3.96 (0–9)
	Mean Pain score [Min–Max]		6.07 (1–10)
	Mean Uterine score ** [Min–Max]		3.51 (0–9)
		Endometriotic nodule: E (*n*, %)	58 (52.2)
		Retroverted uterus: R (*n*, %)	44 (39.6)
		Adenomyosis: A (*n*, %)	27 (24.3)
	Rare score ^†^ (*n*, %)		5 (4.5)

* revised American Society for Reproductive Medicine. ** Uterine Score includes three types of conditions: Endometriotic nodule categorized as ‘E’, retroverted uterus categorized as ‘R’, and adenomyosis categorized as ‘A’. Below is the breakdown of the number and percentage of the cases. ^†^ The Rare Score assigns 10 points to cases with rare-site endometriosis. However, as there were only 5 instances in the current study, its average score was not calculated.

**Table 3 biomedicines-12-01267-t003:** Correlation between various scores, surgical duration, and blood loss.

	r-ASRM Score	E-Score	Cyst Score	Adhesion Score	Pain Score	Uterine Score
r-ASRM score	-	0.758	0.521	0.793	0.223	0.409
Surgery Duration	0.700	0.724	0.323	0.640	0.362	0.491
Blood loss	0.328	0.400	0.201	0.296	0.256	0.267

This table was calculated using the Spearman correlation coefficient method. Note: The numbers indicate the correlation coefficients.

**Table 4 biomedicines-12-01267-t004:** Correlation analysis of surgical duration with combined scoring of adhesion, pain, cyst, Rare, and uterine scores in endometriosis.

	Variation of Additional Element of Uterine Score (US)					
	No Additional Score	+A	+R	+E (All 3) *	+E (3, 6, or 10) **	+E (3, 6, or 10) ** +Rare Score (RS) ^†^
Adhesion score (AS)	0.640	0.606	0.618	0.740	0.745	0.761
AS + Pain score (PS)	0.629	0.642	0.614	0.701	0.720	0.744
Cyst score (CS) + AS + PS	0.641	0.667	0.649	0.713	0.730	0.747
CS + AS + PS + US (A,R,E)	0.713 ^‡^	-	-	-	0.724	*Same as below*
E-score (CS + AS + PS + US (A,R,E) + RS)	0.724 ^‡^	-	-	-	-	0.752 ^§^
r-ASRM score	0.700					

This table was calculated using the Spearman correlation coefficient method. Note: The numbers indicate the correlation coefficients. A: adenomyosis, R: retroverted uterus, E: endometriotic nodules. * 3 points are uniformly allocated for each endometriotic nodule identified by ultrasound or similar diagnostic imaging, regardless of the number of lesions. ** 3 points are allocated for endometriotic nodules sized 1–2 cm, 6 points for those sized 2–3 cm, and 10 points for lesions over 3 cm identified by ultrasound or similar diagnostic imaging. For multiple lesions, scores should be cumulatively added. ^†^ 10 points are added if rare-site endometriosis, such as vaginal endometriosis, is observed. ^‡^ the uterine score has already been added. Endometriotic nodules categorized as ‘E’ are uniformly allocated 3 points each, irrespective of the number of lesions. ^§^ the uterine score has already been incorporated. Endometriotic nodules categorized as ‘E’ have been calculated in a format that assigns 3, 6, or 10 points depending on their size.

## Data Availability

The data supporting the reported results are not publicly available due to privacy and ethical restrictions.

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
