# Peer review of "Feasibility of Predicting Surgical Duration in Endometriosis Using Numerical Multi-Scoring System of Endometriosis (NMS-E)"

_biomedicines, 2024, doi:10.3390/biomedicines12061267_

Round 1

Reviewer 1 Report

Comments and Suggestions for Authors

The authors report on the effectiveness of the Numerical Multi-Scoring System of Endometriosis (NMS-E) in predicting surgical outcomes and correlating with the severity of endometriosis.

They found that a statistically significant positive correlation was observed between the NMS-E’s E-score and the severity of endometriosis, particularly in predicting surgical duration (correlation coefficient: 0.703, p < 0.01) and blood loss (coefficient: 0.407, p < 0.01). The NMS-E E-score also correlated strongly with the r-ASRM scores (coefficient: 0.700, p < 0.01), exhibiting a slightly more excellent predictive value for surgical duration than the revised American Society for Reproductive Medicine (r-ASRM) scores alone.

 Comments

The statistic component of the manuscript is detailed and well conducted.

However, if for a statistician the present report might be useful, I have concerns about the utility for the clinician, the ultimate target of this study, because it is enough complicated to calculate the NMS-E score.

Author Response

Thank you for your thoughtful comments and insights on our manuscript. We will now address your points of concern.

Complexity of Calculating the NMS-E Score:

We acknowledge the complexity involved in calculating the NMS-E score, which could limit its usability in clinical settings. Based on your feedback, we have developed a calculator for the NMS-E score to simplify this process. Additionally, we plan to create a simplified version of the E-SCORE in the future. By selecting key components from the many elements of NMS-E (for example, only the score for E lesions in the Uterine score and adhesion score, as their predictive ability seems to be similar), we aim to make it more accessible and practical for clinicians.

Guidance for Clinicians:

Following your suggestion, we have included the calculator as an appendix, making it accessible to readers. This tool will assist clinicians in interpreting and applying the NMS-E score in their practice, enhancing its clinical utility.

We have also addressed comments from other reviewers. Notably, due to the scores not being normally distributed in our statistics, we changed our statistical methods. This has resulted in slight changes to the numbers in most of the figures and tables.

We greatly appreciate your insightful feedback and the opportunity to enhance our manuscript based on your suggestions. Thank you for your careful consideration and the time you have invested in reviewing our work. Your expertise has been invaluable in guiding our revisions.

Reviewer 2 Report

Comments and Suggestions for Authors

This manuscript is interesting but very difficult to read

In my opinion, there are several problems with it

The proposed score is complicated to understand and in addition it includes many exceptions and corrections that make it very difficult to establish and I doubt that many surgeons will use it in practice.

This score is preoperative, i.e. before laparoscopy, to judge the probable duration of the operation, which is interesting in practice for the scheduling of operating theatres. So why compare it to the AFS score which is only obtained intraoperatively. It would have been more interesting to compare it to a subjective duration given by the surgeon without using this score. In practice, this is what is done in many departments where the surgeon gives a probable duration of the operation, but which is not argued on objective data as in your score

Scores are only used if they are simple to implement or if they are calculated automatically by a small computer program.

This score is then correlated with pain and other symptoms such as dysesechia, dyspareunia, etc. This doesn't add much and ends up complicating this article.

I think that in its current form, very few readers will go all the way to the end of the article. I only went all the way because I was a reviewer

But I am perfectly aware that it took a very large amount of work to write this article.

Author Response

Thank you for your thoughtful comments and insights on our manuscript. We will now address your points of concern.

Complexity of the Proposed Score:

Thank you for your comments. To reduce the complexity of the calculations, we have developed a calculator for the NMS-E score. Additionally, in the future, after gathering more data, we plan to create a simplified version of the E-SCORE by selecting certain elements from the many components of NMS-E (for example, only the adhesion score and the score for E lesions of the uterine score, as their predictive ability appears to be similar). We aim to put this simplified version into practical use.

Comparison to AFS Score:

Indeed, as you pointed out, there is a certain distance between comparing a preoperative scoring method like NMS-E and an intraoperative scoring method like r-ASRM (AFS). Therefore, I have added supplemental data showing a correlation coefficient of 0.932 between preoperative NMS-E data and intraoperative NMS-E data, and a correlation coefficient of 0.803 between intraoperative NMS-E data and r-ASRM (AFS) data. Endometriosis is a complex condition, and predicting surgical duration preoperatively based purely on subjective assessment is quite challenging. Hence, we believe that such a multifaceted and objective evaluation method could be useful as one approach to predicting surgical duration.

Implementation of Scores:

To save the examiner's effort, we have created a calculator for the NMS-E and attached it with the supplementary data. We believe this will be helpful in simplifying the implementation of the score.

Correlation with Symptoms:

To simplify the structure of the paper, the data in this section, as pointed out, has been moved to supplementary materials.

Readability of the Article:

Thank you for your candid feedback. To aid in understanding the paper, we have removed irrelevant data from the text and created a calculator for the NMS-E to facilitate understanding of the score. We hope these adjustments will enhance the comprehension of the article.

Acknowledgement of Effort:

Thank you for acknowledging the effort that went into writing this article. Your recognition means a lot to us.

We greatly appreciate your insightful feedback and the opportunity to enhance our manuscript based on your suggestions. Thank you for your careful consideration and the time you have invested in reviewing our work. Your expertise has been invaluable in guiding our revisions.

Reviewer 3 Report

Comments and Suggestions for Authors

The article's manuscript is devoted to the important and widespread medical problem of endometriosis. I support the enthusiasm and efforts of the authors to improve the numerical multi-scoring system for endometriosis, as it has significant practical implications for the clinic.

My rating is positive, but I have some comments/suggestions.

Statistics play a key role in proving the reliability of the authors' results. Has the normality of the data distribution been assessed? This is not indicated in subsection 2.3 of the Materials and Methods section. This is important for the correct choice of statistical method.

I recommend reworking the "Conclusions" section and filling it with the specific results obtained in this study.

Author Response

Thank you for your thoughtful comments and insights on our manuscript. We will now address your points of concern.

Assessment of Data Normality:

Thank you for your comments. As you noted, the initial manuscript did not address the evaluation of data normality. Following your suggestion, I performed a Shapiro-Wilk test, which confirmed that the data do not follow a normal distribution, with a W statistic of 0.97205 and a p-value of 0.01966. Consequently, I have revised the statistical analysis using non-parametric Spearman methods, which are suitable for non-normally distributed data, and updated the manuscript accordingly.

Reworking the Conclusions Section:

I have revised the 'Conclusions' section as suggested, incorporating the specific results obtained in our study.

Additionally, we have addressed comments from other reviewers as well. Due to the scores not being normally distributed, we changed our statistical methods, which resulted in slight changes to the numbers in most of the figures and tables.

We greatly appreciate your insightful feedback and the opportunity to enhance our manuscript based on your suggestions. Thank you for your careful consideration and the time you have invested in reviewing our work. Your expertise has been invaluable in guiding our revisions.

Reviewer 4 Report

Comments and Suggestions for Authors

Comments on the manuscript where the authors propose a new numerical scoring evaluation system to improve the diagnosis of endometriosis to improve surgical planning and patient management. Some suggestions are listed below.

Introduction paragraph 1 lines 47-50, and 59, 71, 79, 101 include reference

Line 67, 74 meaning of #

Define abbreviations used in the introduction

Fig. 1 can be omitted and only Figure 2 included.

Point 2.2.2. It seems that the lyrics are incomplete

Check for typographical errors (punctuation)

Include tables 1 and 2 in the results text

Tables 1 and 2 eliminate Mean (Min-Max) n (%) from the first row since they are defined in the rows in the left column of the table.

Table 1 Define y.o., align the rows of Table 1 and 2, and in the rows of Dysmenorrhoea (n=102*) Mean VAS value 6.8, define the subsequent ones with Mean VAS value if this is the case (Dyspareunia, Dyschezia, and Chronic pelvic pain)

Strengthen the discussion with other related studies (lines 378-424, 435-453)

Homogenize the writing of references according to the journal instructions

Author Response

Thank you for your thoughtful comments and insights on our manuscript. We will now address your points of concern.

Introduction Paragraph References:

I have made the necessary adjustments and included the references in the specified lines.

Meaning of # in Lines 67 and 74:

Thank you for your question regarding the use of the "#" symbol in the term "#Enzian." It's important to clarify that the "#" is an integral part of the official name of this diagnostic method and not an addition or typographical error. Originally, the symbol was presented in superscript, which might have led to some confusion. To improve readability and avoid any misunderstanding, we have updated the format by changing the superscript "#" to a regular, in-line symbol while ensuring it remains as part of the official name.

Define Abbreviations Used in the Introduction:

I have made the necessary adjustments according to the comment.

Omission of Figure 1:

Per your advice, Fig. 1 has been removed from the main text and included as a Supplementary Figure for reference by interested readers.

Completion of Section 2.2.2:

Thank you for your comment regarding Section 2.2.2. In response, I have reorganized Section 2.2.2 to clarify the application of NMS-E, and I have added a new Section 2.2.3 to include detailed information on the calculation method for the E-score.

Typographical Errors (Punctuation):

Thank you for your feedback regarding typographical errors, particularly concerning punctuation. We will have the document reviewed by a native speaker to ensure that all punctuation is correctly used and that any typographical errors are addressed. This will help enhance the clarity and readability of the text.

Inclusion of Tables 1 and 2 in the Results Text:

Thank you for your suggestion. I have now included Tables 1 and 2 within the results section of the text as advised.

Elimination of Mean (Min-Max) n (%) from First Row of Tables 1 and 2:

Thank you for your feedback. I have removed the "Mean (Min-Max) n (%)" from the first row of Tables 1 and 2, as these details are already defined in the rows in the left column of each table as suggested.

Definition and Alignment in Table 1:

I have defined "y.o." and aligned the rows of Tables 1 and 2. For the rows of Dysmenorrhoea (n=102*) Mean VAS value 6.8, I have also defined the subsequent ones with Mean VAS value if this is the case (Dyspareunia, Dyschezia, and Chronic pelvic pain).

Strengthening the Discussion with Related Studies:

I have strengthened the discussion section with additional related studies as suggested.

Homogenizing the Writing of References:

I have revised the references to adhere to the 'ACS' style according to the journal instructions.

We greatly appreciate your insightful feedback and the opportunity to enhance our manuscript based on your suggestions. Thank you for your careful consideration and the time you have invested in reviewing our work. Your expertise has been invaluable in guiding our revisions.

Round 2

Reviewer 1 Report

Comments and Suggestions for Authors

The authors answered to my concerns. 

Reviewer 2 Report

Comments and Suggestions for Authors

This article is very special and particularly complex to review.

Based on a preoperative, clinical and ultrasound score, the authors try to correlate this score with the operative time and the duration of the procedure.

This  research for a score to replace the AFS score (intra-operative) is commendable and remains a current issue, but much less for the operative duration than for other more interesting data such as the chance of pregnancy in the event of infertility or the improvement of pain after surgery

This study has some major limitations : it is retrospective, unicentric and even with a score established by a single operator

The score used is complex to implement and must be clarified because there are too many exceptions difficult to interstand

On the other hand, the effort made for such a study is obvious and this line of research must be continued

- by prospective studies

- by multicenter and multi-operator studies

- with a simplification of the score

- and by considering other parameters such as fertility and the disappearance of pain after surgery

Also, my suggestion is to accept this article by simply modifying the title to show the very preliminary nature of this study and for example by adding: "preliminary report on " before the current tittle